# OpenReview forum: "Unsolvable Problem Detection: Evaluating Trustworthiness of Large Multimodal Models"
_ICLR.cc/2025/Conference — Submitted to ICLR 2025_

### Official Review · Reviewer_48si · 2024-11-01

**Soundness:** 2
**Presentation:** 2
**Contribution:** 2
**Rating:** 6
**Confidence:** 2

**Summary:**

This paper introduces a new challenge for LMMs, termed Unsolvable Problem Detection (UPD) which examines the LMM’s ability to withhold answers when faced with unsolvable problems. Then, a benchmark is introduced to assess the performance across various ability dimensions and some solutions are explored to understand UPD. It can enhance the development of more practical and reliable LMMs.

**Strengths:**

It first introduces a new challenge for LMMs, termed Unsolvable Problem Detection (UPD).
It constructs a benchmark to assess the performance across various ability dimensions.
It evaluates state-of-the-art LMMs on the UPD problem with the benchmark.

**Weaknesses:**

The dataset in the paper is constructed based on existing datasets, and there is no comparative analysis of data size, distribution, and comprehensiveness, etc.

**Questions:**

What is the difference between the exsiting definition of unsolvable problems and the definition in this paper?
What is the difference in the design of Table 3 and Table 4, and why not merge them？

---

> ### Author Response · Authors · 2024-11-20
> **Response to Weaknesses 1-2**
>
> We sincerely appreciate your time and effort in reviewing our work. Below, we provide detailed responses to each of your points. If you have other questions, we welcome further discussion!
>
> *Note: Unless otherwise specified, the mentioned line numbers denote that of the original submitted paper version instead of the rebuttal revision version.*
>
> > ### **W1. Difference in the Definition of Unsolvable Problem**
>
> Thank you for your question. The differences from existing settings [1, 2] are described in the Introduction (L64-66). While existing benchmarks primarily address mismatches between images and questions (a small partition of MM-IVQD’ abilities), they have overlooked other critical challenges such as incomplete or missing answer sets (AAD and IASD). To expand the scope of unsolvable problems, we have rigorously defined three types: AAD, IASD, and IVQD. We have also introduced the MM-UPD Bench, which evaluates all these unsolvable problems across various abilities. Our experimental results reveal performance differences among LMMs for each unsolvable problem type (Finding 5, L412-422), underscoring the necessity of comprehensive coverage for all types.
>
> > ### **W2. Differences in Design Between Tables 3 and 4 and Rationale for Keeping Them Separate**
>
> Thank you for the question. Both Table 3 and Table 4 aim to explore baseline approaches for UPD. The reason we do not merge them is that there is a fundamental difference in the approaches themselves. Table 3 focuses on prompting-based approaches, while Table 4 deals with training-based approaches.
>
> **References**
>
> [1]  Guo+, UNK-VQA: A Dataset and a Probe into the Abstention Ability of Multi-modal Large Models, TPAMI2024
>
> [2] Akter+, VISREAS: Complex Visual Reasoning with Unanswerable Questions, ACL Findings2024

---

> > ### Author Response · Authors · 2024-11-25
> > **Gentle Reminder to Reviewer 48si**
> >
> > Dear Reviewer 48si,
> >
> > We appreciate all the constructive comments. Your feedback has been extremely helpful in improving our paper. Based on your comments, we have written our response.
> >
> > As the discussion period ends soon, we would greatly appreciate your feedback or any follow-up questions to confirm whether our responses have sufficiently addressed your concerns. We would be happy to provide any further details to answer your follow-up questions.
> >
> > Warm regards, Authors

---

> > > ### Author Response · Authors · 2024-12-01
> > > **Response Reminder**
> > >
> > > Dear Reviewer 48si,
> > >
> > > As the discussion period is coming to a close (only 2 days remaining), we would greatly appreciate the feedback on whether we have sufficiently addressed your concerns. If the reviewer has any further concerns, we would be happy to address them.
> > >
> > > Best regards, Authors

---

### Official Review · Reviewer_A2BT · 2024-11-04

**Soundness:** 2
**Presentation:** 3
**Contribution:** 2
**Rating:** 5
**Confidence:** 4

**Summary:**

This paper addresses a challenge in multimodal large language models (MLLMs): Unsolvable Problem Detection (UPD). The authors conceptualize UPD through three distinct tasks: Absent Answer Detection (AAD), Incompatible Answer Set Detection (IASD), and Incompatible Visual Question Detection (IVQD). To systematically evaluate MLLMs’ capabilities in addressing UPD, they introduce a benchmark, MM-UPD (derived from MMBench), and assess both open-source and closed-source models on this benchmark. The evaluation reveals that current models exhibit substantial difficulty in handling UPD tasks. To mitigate these limitations, the authors propose a series of enhancement strategies, including chain-of-thought prompting, self-reflection, and instruction tuning, aimed at improving MLLMs' effectiveness in detecting unsolvable problems.

**Strengths:**

1. The authors investigate the important issue of Unsolvable Problem Detection (UPD) and provide an in-depth analysis of this challenge within the context of multimodal large language models (MLLMs).

2. To evaluate MLLMs' effectiveness in handling UPD, they develop a benchmark that incorporates diverse evaluation settings, examining various scenarios and testing a range of both open-source and closed-source MLLMs. The findings, particularly those outlined in Section 5.2, offer valuable insights into current MLLMs limitations.

3. To address the UPD challenge, the authors propose several practical strategies, including chain-of-thought prompting (CoT), self-reflection, and instruction tuning, which show promise in mitigating the difficulties MLLMs face in detecting unsolvable problems.

**Weaknesses:**

1. **Narrow Definition of Unresolvable Problem Detection**: The proposed definition of Unresolvable Problem Detection (UPD) is limited, focusing solely on multiple-choice problems. This approach overlooks other open-ended multimodal question-answering scenarios, which are prevalent and significant in real-world applications. Expanding the scope to include these scenarios would provide a more comprehensive understanding of UPD.

2. **Anticipated Poor Performance of Existing MLLMs**: The observed poor performance of existing multimodal large language models (MLLMs) in handling UPD is not unexpected. This is likely due to the absence of specific instruction-tuning datasets during the visual instruction fine-tuning stage. While the proposed solutions, such as chain-of-thought prompting, self-reflection, and instruction tuning, are steps in the right direction, they appear somewhat straightforward. More sophisticated and innovative approaches are needed to effectively address the complexities of UPD.

3. **Limited Benchmark Diversity**: The benchmark used for evaluating MLLMs' capability in handling UPD lacks diversity, relying solely on MMBench, which comprises 2,095 questions. This limited dataset may not provide a comprehensive evaluation. The exclusion of other datasets like MMMU and MathVista, based on their low standard accuracy, may not be a sufficiently robust justification. Incorporating a variety of datasets would enhance the evaluation's robustness and offer a more thorough assessment of MLLMs' performance in UPD tasks.

**Questions:**

1. **Manual Dataset Curation Standards**: The benchmark dataset involves certain manual efforts, such as eliminating image-agnostic questions and removing ambiguous questions. Could the authors clarify the criteria used for these removals? What standards were applied to ensure consistency and objectivity in these decisions?

2. **Alternative Solutions for UPD**: Are there potentially more advanced or effective solutions for addressing the challenges of Unsolvable Problem Detection (UPD) beyond those proposed? If so, what approaches could further improve MLLMs' performance in handling UPD scenarios?

3. The error analyses in section 6.2 is insightful, could you do the same analyses for open-source MLLMs?

---

> ### Author Response · Authors · 2024-11-20
> **Response to Weaknesses 1-2**
>
> We sincerely thank you for your thorough review of our paper and your constructive feedback on how we can further advance our UPD challenge.  We are particularly grateful that you acknowledge our strengths, especially  **“S2. The findings offer valuable insights into current MLLM’s limitations”**. We have carefully considered your suggestions and updated the manuscript accordingly, with changes highlighted in orange. Below, we provide detailed responses to each of your points:
>
> *Note: Unless otherwise specified, the mentioned line numbers denote that of the original submitted paper version instead of the rebuttal revision version.*
>
> > ### **W1. Narrow Definition of Unresolvable Problem Detection**
>
> We appreciate your critical question. MM-UPD enables more accurate evaluation than open-ended questions, and **we believe that conducting thorough and precise evaluation using multiple-choice questions is a crucial initial step in developing reliable LMMs.** While we agree that open-ended questions are important, their evaluations are inherently challenging. Therefore, most established benchmarks use multiple-choice questions, including MMBench, MMMU [1], BLINK [2], and MMStar [3]. However, the problem with evaluating multiple-choice questions is that it is unclear whether the model truly understands the question or is just using a process of elimination, which undermines the guarantee of reliability. We believe that the multiple-choice UPD enables precise evaluation and reveals the reliability of LMMs, which inspires future efforts in this field. We have clarified this description in the revised Future Work section (Section 7).
>
> > ### **W2.1 Anticipated Poor Performance of Existing MLLMs**
>
> We would like to point out that our findings go beyond mere performance degradation and reveal critical insights that can only be obtained through our rigorous task definition and benchmark construction. For instance, little correlation with MMBench (Table 2), the differences in the effectiveness of prompting strategy (F4, **L402-411**), performance trends of AAD/IASD/IVQD across MLLMs (F5, **L412-422**), and the variations in trends across abilities (F6, **L422-431)** could not have been anticipated without a meticulous problem definition, benchmark design, and establishment of evaluation metrics. We consider that these findings are not trivial and cannot be predicted without this paper.
>
> > ### **W2.2 Lack of Innovative Approaches**
>
> **We would like to emphasize that this study primarily focuses on the rigorous task design of UPD and proposing approaches is not the main contribution of this paper.** For a novel task, it is essential to demonstrate the effectiveness and limitations of existing important baselines before proposing innovative methods. As **Reviewer gvew** also pointed out, we believe our exploration of important baselines will accelerate the community's efforts to propose innovative approaches. We have clarified the description in the Future Work Section 7.

---

> ### Author Response · Authors · 2024-11-20
> **Response to Weakness 3**
>
> > ### **W3.1 Limited Benchmark Diversity: Validity of Using only MMBench**
>
> Thank you for your question. We consider that using only MMBench does not result in a limited dataset. MMBench is a benchmark specifically designed to objectively and systematically evaluate the diverse capabilities of LMMs, and it has been widely adopted as a reliable evaluation tool in numerous research papers.
>
> In terms of dataset scale, MM-UPD includes 2,095 questions, which is comparable to other recent multi-choice benchmarks: MMMU validation (widely used in academic papers) [1] with 900 questions, BLINK [2] with 3,807 questions, MMStar [3] with 1,500 questions, and MMMU-Pro (vision) [4] with 1,730 questions.
>
> Furthermore, we believe that the sufficiency of a dataset scale is better judged by its ability to produce meaningful insights. In this regard, the evaluation results of MM-UPD have already provided significant findings. Based on these considerations, we consider that MM-UPD offers a dataset of sufficient scale to serve as a reliable benchmark for UPD.
>
> > ### **W3.2** **Limited Benchmark Diversity: Rationale for Excluding Datasets with Low Standard Accuracy**
>
> Thank you for this important question. The reason is that **evaluation results from benchmarks with low Standard Accuracy could significantly deviate from the aspect of reliability and potentially cause us to miss important findings.**  We have added these discussions in the revised Appendix B.6.
>
> To verify this, we conducted experiments with MMMU in the AAD setting.
>
> Evaluation Setup:  As preprocessing, we first removed about 24.2% of image-agnostic questions from the MMMU's validation set (900 questions) using GPT-4-based CircularEval. Then, to improve the interpretability of scores, we utilized only multiple-choice questions with four options (which make up the majority of problems in MMMU) and created MMMU-AAD using the same pipeline as in the paper. MMMU-AAD consists of 459 problems. For the evaluation of MMMU-AAD, we applied the same CircularEval strategy as used in MM-UPD.
>
> | LMM | Orig. | Base (Standard/UPD) | Option (Standard/UPD) | Instruction (Standard/UPD) |
> | --- | --- | --- | --- | --- |
> | LLaVA-OV-7B | 23.5 | 0.7 (20.5, 5.7) |  0.7 (22.4/2.4) | 0.7 (20.0/2.4) |
> | InternVL2-8B | 24.4 |  4.1 (19.8, 9.4) | 2.8 (22.0, 4.1) | 3.5 (21.8, 11.8) |
> | LLaVA-NeXT-34 | 23.9    | 6.3 (12.0, 35.4) | 0.4   (23.4, 1.8) | 4.2 (9.6,  59.7) |
> | GPT-4o | 27.5※ | 15.5 (42.9,   20.9) | 8.9 (24.4, 19.0)　 | 23.7 (35.9, 48.4) |
>
> ---
>
> ※ The reason GPT-4o's Original Standard performance is lower than its Base Standard is that GPT-4o generates extensive long reasoning for challenging datasets like MMMU, solving problems with a chain-of-thought process. However, this arises from GPT-4o's proprietary tuning strategy and this is unrelated to UPD. Therefore, we omit it from our discussion here.
>
> ---
>
> Based on these results, in contrast to MM-UPD, we could not verify the efficacy of either the Option or Instruction approaches. **This result reveals that the evaluation using MMMU fails to capture important findings of the effectiveness of these prompting approaches for UPD.** Specifically, for expert-level problems, MLLMs do not have accurate answers due to the lack of capability. Therefore, even if it chooses an incorrect option when encountering an unsolvable problem, this only indicates a lack of reasoning ability or knowledge and does not necessarily demonstrate a lack of refusal ability. Additionally, due to the very low overall performance, it becomes difficult to have meaningful discussions based on these minute differences in scores. Therefore, we exclude datasets with low Standard accuracy.  However, as the model's capabilities improve much more, it will become increasingly important to evaluate from multiple perspectives using more complex datasets, as pointed out by the reviewer. We consider this will be a challenge for future research.

---

> ### Author Response · Authors · 2024-11-20
> **Reesponse to Questions 1-3**
>
> > ### **Q1. Manual Dataset Curation Standards**
>
> Thank you for pointing out. We have added the details of the curation procedure to the revised Appendix B.5.
>
> The dataset curation was primarily conducted by four annotators among the authors.
>
> - To improve the efficiency of collaborative curation and ensure consistency in quality, we first transcribed the image-question pairs from MMBench into an online editing tool (i.e., Google Docs) and conducted the curation process directly within the platform.
> - To enhance the consistency, each question was independently reviewed by two annotators. Finally, the lead author verified the validity of all curation.
> - If a problem needed to be refined, the reason was recorded in detail as a comment. For example, in the case of IVQD, which required the most careful curation, one annotator would leave a comment on points such as “The reason the image relates to the question is...” or “If we change this image into ..., the irrelevance is guaranteed". If the other annotator agreed with the comment, the problem was refined. In cases where the other annotator disagreed, all four annotators engaged in discussions to reach a consensus.
>
> We consider that collaborative tools such as Google Docs, double-checking by two annotators, and detailed justifications with collective decisions ensure curation consistency.
>
> > ###  **Q2. Alternative Potential Solutions for UPD**
>
> We consider that one of the promising directions is to mimic the human reasoning process with chain of thought. When humans identify flaws in a problem, they first derive their own answer and then compare it to the problem. By replicating this process, MLLMs may be able to prevent blindly outputting incorrect answers. However, the development and analysis of this new chain of thought techniques clearly goes beyond the scope of this paper. Therefore we position it as a topic for future research.
>
> > ### **Q3. Analysis of Open-source MLLMs**
>
> Thanks for the suggestion. We have added the same analysis for open-source MLLMs in the revised Appendix E.1. We observe that InternVL2 showed high performance, and the bottleneck lies in image understanding. On the other hand, LLaVA-OV, LLaVA-NeXT, and Qwen2VL showed low performance even when given answers, and the bottleneck was found to be on the language side. This analysis result supports our finding in Table 3 that LLM-driven approaches like chain of thought and self-reflection are particularly effective for LLaVA-OV and LLaVA-NeXT. We consider these will provide valuable insights for the open-source community to develop more reliable models.
>
> **References**
>
> [1] Yue+, MMMU: A Massive Multi-discipline Multimodal Understanding and Reasoning Benchmark for Expert AGI, CVPR2024
>
> [2] Fu+, BLINK: Multimodal Large Language Models Can See but Not Perceive, ECCV2024
>
> [3] Chen+, Are We on the Right Way for Evaluating Large Vision-Language Models?, NeurIPS2024
>
> [4] Yue+, MMMU-Pro: A More Robust Multi-discipline Multimodal Understanding Benchmark, arXiv2024

---

> > ### Author Response · Authors · 2024-11-25
> > **Gentle Reminder to Reviewer A2BT**
> >
> > Dear Reviewer A2BT,
> >
> > We appreciate all the constructive comments. Your feedback has been extremely helpful in improving our paper. Based on your comments, we have written our response and revised the manuscript.
> >
> > As the discussion period ends soon, we would greatly appreciate your feedback or any follow-up questions to confirm whether these revisions have sufficiently addressed your concerns. We would be happy to provide any further details to answer your follow-up questions.
> >
> > Warm regards, Authors

---

> > > ### Author Response · Authors · 2024-12-01
> > > **Response Reminder**
> > >
> > > Dear Reviewer A2BT,
> > >
> > > As the discussion period is coming to a close (only 2 days remaining), we would greatly appreciate the feedback on whether we have sufficiently addressed your concerns. If the reviewer has any further concerns, we would be happy to address them.
> > >
> > > Best regards, Authors

---

> > > > ### Comment · Reviewer_A2BT · 2024-12-02
> > > > **response**
> > > >
> > > > Thank you for the author's response. While the work primarily focuses on the rigorous task design of UPD, it lacks comprehensive analysis and fails to provide a more generic task definition, which are important for future work to build upon. As such, I will maintain my original score.

---

> ### Author Response · Authors · 2024-12-03
> **Response to review A2BT**
>
> We sincerely appreciate your thoughtful feedback and valuable insights.
>
> As stated in the Future Work section, we acknowledge the importance of addressing more general tasks, including open-ended questions.
>
> Nevertheless, our multiple-choice UPD serves a valuable purpose—**it introduces a novel dimension of reliability assessment within established multiple-choice formats and offers an effective way to measure robust understanding**.
>
> Through our carefully designed problem framework, we have demonstrated that models exhibiting poor performance on UPD tasks indicate a lack of prediction reliability, while those performing well demonstrate robust understanding and high reliability. Although many established benchmarks adopt multiple-choice formats [1, 2, 3, 4], they lack methods to assess this depth of understanding.
>
> Furthermore, our benchmark, based on MMBench, accurately evaluates the refusal capability axis, providing precise feedback to enhance reliability (refer to the response to W3.2).
>
> Through UPD’s systematic design and rigorous evaluation framework, we believe our UPD can advance efforts to improve LMM reliability.

---

### Official Review · Reviewer_gvew · 2024-11-04

**Soundness:** 3
**Presentation:** 3
**Contribution:** 3
**Rating:** 6
**Confidence:** 4

**Summary:**

The paper introduces Unsolvable Problem Detection (UPD) as a framework to evaluate Large Multimodal Models (LMMs) on their capacity to identify and manage unsolvable problems. This framework consists of three main challenges: Absent Answer Detection (AAD), Incompatible Answer Set Detection (IASD), and Incompatible Visual Question Detection (IVQD), to address scenarios involving insufficient, mismatched, or incompatible information. To support evaluation, the authors present the MM-UPD Bench, a benchmark consisting of 2k questions derived as a subset of MMBench, to measure LMM performance in these areas. The paper evaluates several LMMs using the MM-UPD Bench and provides an analysis based on open-source versus closed-source models, as well as detailed question types.

The experimental results reveal that even state-of-the-art LMMs, which perform well on conventional benchmarks, encounter difficulties when assessed with the MM-UPD Bench. Additionally, generic existing approaches were tested to address UPD but proved ineffective, which underscores the challenge of UPD.

**Strengths:**

1. The paper is well-written and easy to follow. The definitions of the problems and evaluation criteria are clearly presented, and the visualizations of examples and results enhance the readers' understanding.
2. The paper conducts comprehensive and detailed experiments to analyze where and how LMMs struggle with UPD. It further evaluates the effectiveness of existing methods in addressing UPD, which offers valuable insights for future researchers aiming to tackle these challenges.

**Weaknesses:**

1. The paper's strategy for filtering image-agnostic questions (line 207) to ensure that problems truly rely on the image modality seems problematic. Specifically, the method filters questions from MMBench that can be answered correctly using text-only GPT-4 and manually reviews the remaining questions. This approach risks excluding genuinely image-dependent questions that GPT-4 might answer correctly by chance. As shown in Table 1 IVQD, there is a possibility that GPT based model attempts to answer questions that require image information but lack relevant image input. This could lead to false exclusions, and introduce bias into the dataset and impact the evaluation of GPT-based models.
2. Following the previous point, the paper states in line 996 that 13% of the original questions are removed as image-agnostic. This suggests that the initial GPT-based filtering only marginally reduces manual effort. Given this, why not manually filter all image-agnostic questions to ensure greater accuracy and consistency?
3. The use of Dual Accuracy as the primary metric (line 252) may not fully capture the intended evaluation properties of the benchmark. Dual Accuracy is defined as *the accuracy on standard-UPD pairs, where success is counted only if the model answers both the standard and UPD questions correctly.* However, as illustrated in Figure 2's examples, a model might correctly refuse to answer absurd questions even if it is not capable of answering the standard version. This coupling of general VQA ability and the capacity to reject unanswerable questions could skew comparisons across models. To better assess the models' capability in detecting unsolvable questions, it may be more informative to compare success rates on questions where the model correctly answers the standard version. This ensures both text and image are understood, and evaluates if the changes that make the question becomes unanswerable are detected. (Mathematically, it's similar to divide Dual Accuracy by Standard Accuracy).
4. The distribution of question types appears skewed and inconsistent across AAD, IASD, and IVQD, as shown in Table B (line 1042). For instance, in AAD and IASD, the three most common question types (#2, #17, and #3) account for 30% of the questions, whereas in IVQD, the most common types (#2, #7, and #9) account for 44%. Could this affect the analysis of UPD performance? Additionally, six question types have no representation in IVQD. While it is understandable that certain types need to be removed to evaluate IVQD, if the distribution matters (as stated in the previous point), could it be more beneficial to modify these questions and make them less general rather than remove them?
5. Table 4 reports performance following UPD-specific training. The paper notes that such training may degrade performance on general tasks. However, the performance under the *Orig* condition remains the same as indicated in Table 1. To provide a complete analysis, it would be helpful to re-evaluate the *Orig* condition after fine-tuning to observe any potential impacts.

**Minor Points:**

1. Fix the typo: *withcurrent* (line 135).
2. Consider reformatting Figure 2 and increasing the text font size for improved readability.

**Questions:**

1. Please refer to the concerns raised in Weaknesses 2 and 4.

2. In addition to the three challenges defined in the paper, have the author considered an additional challenge of *incompatible question detection*? This would involve scenarios where the image and answer set align, but the question itself is incompatible with both the image and the answer. For example, replacing the original question in an image-question-answer set with an unrelated one. This condition appears aligned with how the proposed challenges are structured. Would it be necessary to incorporate this condition, or is this concept already encompassed by the current challenges defined?

---

> ### Author Response · Authors · 2024-11-20
> **Response to Weaknesses 1-3**
>
> We sincerely thank you for your insightful and constructive feedback. Your comments are highly valued and critical to help make our paper more robust. We are particularly grateful that you correctly understood the strengths of this paper, specifically noting **"S2. comprehensive experiments and analysis"**, and **"S.2. Insightful evaluation of key baselines methods"**.
>
> We have carefully considered your suggestions and updated the manuscript accordingly, with changes highlighted in magenta. Below, we provide detailed responses to each of your points:
>
> *Note: Unless otherwise specified, the mentioned line numbers denote that of the original submitted paper version instead of the rebuttal revision version.*
>
> > ### **W1. False Exclusions and GPT-derived Bias in Image-agnostic Question Filtering**
>
> We appreciate your insightful question on our filtering process. **We conducted an investigation and confirmed that our filtering process does not introduce bias in the evaluation of GPT-based models.** We have added the following detailed filtering process and the investigation in Appendix B.1.
>
> To eliminate the effect of random guessing, we applied CircularEval, which has proven effective in eliminating random guessing in MMBench, for filtering. As L307-308, CircularEval cyclically shifts the answer options for each question and considers a prediction a success only if GPT-4 correctly identifies the answer for all shifts. Only 11% (124/1164) of the questions were excluded, which is significantly lower than the rate of random guessing (approximately 30%). This indicates that erroneous exclusions due to random guessing were minimized.
>
> Additionally, to investigate GPT-based biases, we thoroughly examined all the 124 questions excluded by GPT-4. As a result, we found that 110 of 124 were questions that could be answered using only the question texts. The remaining 14 questions appeared image-specific but could be answered by GPT-4 using information from its training, such as the frequency of words in the answer options. However, these 14 questions were primarily limited to a small portion of common questions such as "Which Python code can generate the content of the image?" in #13 or "Which term matches the picture?" Therefore, the impact of removing these 14 questions is considered to be negligible. We confirmed that our MM-UPD does not have GPT-4 bias.
>
> > ### **W2. Effect of GPT-based Filtering on Manual Effort**
>
> Thanks for the question. **We consider that GPT-4-based filtering allows for efficient and precise removal of image-agnostic questions.** Although we needed to manually review the remaining dataset, initial GPT filtering efficiently reduced oversights compared to checking all questions by hand. Our manual double-checks revealed that GPT-4 with CircularEval performed well. The few questions that GPT-4 could not eliminate were mostly limited to the query on direction (e.g., "What direction is France in the Mediterranean Sea? A: east, B: south, C: west, D: north"). Therefore, manual removals require minimal effort during our final checks.
>
> > ### **W3. Dual Accuracy's Limitations in Fair Model Comparison**
>
> Thank you for your insightful suggestion. **We have added experiments comparing the metrics suggested by the Reviewer and Dual accuracy, and we have confirmed that Dual accuracy is effective for the evaluation.** The following discussion has been added in the revised Appendix D.1
>
> We interpret the evaluation metric proposed by the reviewer as the Original Conditional Dual accuracy (OC-Dual) score:  OC-Dual = (Success in all Original Standard, Standard, UPD settings) / (Success in Original Standard)**.** If we use Standard instead of Original Standard as the denominator, models like CogVLM with instruction (Standard: 5.4%, UPD: 92.3% from Table H) that reject most questions could achieve high SC-Dual scores. Therefore, we used the Original Standard as the denominator.
>
> The analysis in the revised paper revealed a very strong correlation between the two metrics. This is attributed to the fact that the Original Standard performance of current LMMs shows little variation within the MM-UPD Bench. The OC-Dual score considers the performance under the successes in the original setting. Therefore, even LMMs with very low Original Standard performance might achieve high scores,  potentially leading to a gap between the score and practical usability. Given the weakness of OD-Dual, using the Dual accuracy for MM-UPD is the most effective to precisely assess the reliability of state-of-the-art LMMs without compromising real-world applicability.

---

> ### Author Response · Authors · 2024-11-20
> **Response to Weaknesses 4-5, Question, Minor 1-2**
>
> > ### **W4. Skewed Distribution of MM-IVQD**
>
> Thank you for your important question. **We consider the current distribution of IVQD questions naturally emphasizes abilities where IVQD situations are more likely to occur, reflecting a natural skew in the IVQD setting.** We have added the following discussion in Appendix B.4.
>
> To align with the AAD/IASD distribution, it is possible to increase the number of questions by making the general question (e.g.,  "Which one is the correct caption of this image?" in #15 Image Topic or "What will happen next?" in #14 Future Prediction) more specific. However, these question types are inherently less likely to encounter IVQD situations, and there is a concern that forcibly modifying the questions might lead to a divergence from real-world IVQD distribution. Moreover, incorporating numerous question types with low IVQD frequency could overshadow the significance of question types that are more likely to occur, thereby compromising the accurate assessment of IVQD performance. Therefore, we chose to exclude these questions rather than modify them.
>
> > ### **W5. Lack of the Original Performance Report after Training**
>
> Thanks for pointing it out. We have added the Original Standard accuracy after training to the table in the updated manuscript. The score is the following.
>
> LLaVA-NeXT-34B
> | LMM | Orig (before training) | Orig (after training) |
> | --- | --- | --- |
> | AAD | 84.3 | 78.6 |
> | IASD | 80.2 | 74.8 |
> | IVQD | 80.9 | 74.7 |
>
> LLaVA-NeXT-13B
> | LMM | Orig (before training) | Orig (after training) |
> | --- | --- | --- |
> | AAD | 76.7 | 68.9 |
> | IASD | 73.2 | 65.4 |
> | IVQD | 71.3 | 67.4 |
>
> The result shows that the Original Standard performance has decreased after training. Developing a training recipe that maintains the Original Standard performance while increasing refusal ability is a challenge for future work.
>
>
> > ### **Q. About Incompatible Question Detection**
> >
>
> Thanks for the interesting question. The primary reason for not incorporating this setup is that the cases where the image and answer set are aligned are limited compared to the three challenges defined in our paper. We consider that ensuring a state where the image and answer set are aligned is challenging and such alignment is primarily limited to specific tasks like image captioning. In most questions, the scenario involves a misalignment between image and answer sets, which can be interpreted as a combination of IASD and IVQD. We judged that analyzing the findings from the combined setting of IASD and IVQD would result in findings similar to those of individual IASD and IVQD. Therefore, considering the practical infrequency of such scenarios and the likelihood of similar findings, we decided not to define this problem setting as a UPD task.
>
> > ### **Minor 1.  Fix the typo: *withcurrent* (line 135).**
> >
>
> Thanks for pointing out. We have fixed this typo.
>
> > ### **Minor 2.  Consider reformatting Figure 2 and increasing the text font size for improved readability.**
> >
>
> Thank you for pointing out. We will reformat Figure 2 and increase the text font size in the final version.

---

> > ### Author Response · Authors · 2024-11-25
> > **Gentle Reminder to Reviewer gvew**
> >
> > Dear Reviewer gvew,
> >
> > We appreciate all the constructive comments. Your feedback has been extremely helpful in improving our paper. Based on your comments, we have written our response and revised the manuscript.
> >
> > As the discussion period ends soon, we would greatly appreciate your feedback or any follow-up questions to confirm whether these revisions have sufficiently addressed your concerns. We would be happy to provide any further details to answer your follow-up questions.
> >
> > Warm regards, Authors

---

> > > ### Comment · Reviewer_gvew · 2024-11-26
> > >
> > > Thank you to the authors for the detailed response and the effort put into the revisions. My concerns have been addressed, and I have adjusted my rating accordingly. I also agree with reviewer A2BT that incorporating a broader scope of UPD questions, such as open-ended questions, could be valuable. An exploratory analysis of how a model performs on the proposed benchmark compared to real-world UPD scenarios would provide further insights.

---

> > > > ### Author Response · Authors · 2024-11-26
> > > > **Official Comment by Authors**
> > > >
> > > > Dear Reviewer gvew,
> > > >
> > > > Thank you once again for your valuable feedback and updated review! We greatly appreciate your positive recommendation.
> > > >
> > > > We also agree with the reviewers' (A2BT and gvew) opinion that it would be more valuable to cover more types of UPD.
> > > > We believe that our findings serve as a foundational first step in this field, paving the way for future efforts to cover a broader range of UPD types.
> > > >
> > > > Best regards, Authors

---

### Author Response · Authors · 2024-12-04
**Discussion Summary**

We sincerely appreciate the constructive discussion. We would like to provide a summary of the discussion phase to offer greater clarity.

The three reviewers generally recognized the strengths of our paper as follows:

- **Reviewer gvew:** "S2. Comprehensive experiments and analysis", "S.2. Insightful evaluation of key baselines methods”, and “S1. The presentation is good and easy to follow.”
- **Reviewer A2BT:** “S1. In-depth analysis of the UPD challenge”, “S2. The findings offer valuable insights into current MLLM’s limitations” and  “S3. Several practical strategies show promise for unsolvable problems.”
- **Reviewer 48si:** “A benchmark to assess the performance across various ability dimensions”

While the reviewers recognized the above strengths, they also expressed some concerns and questions:

- **Reviewer gvew:  Feedback for making this paper more robust.** The reviewer raised concerns about the validity of the filtering process in dataset construction, the appropriateness of data distribution, and the validity of evaluation metrics.
- **Reviewer A2BT: Feedback about how UPD could be advanced further.** The reviewer raised concerns about the coverage of the UPD type, the novelty of the methodology, and benchmark diversity.
- **Reviewer 48si: Questions to clarify differences from existing research.** The reviewer raised questions about the difference in the definition of unsolvable problems from existing ones and concerns about table composition.

During the discussion phase, we diligently and thoroughly addressed each reviewer’s concerns.

- **To Reviewer gvew:**

    (i) **Dataset Construction**: We added more detailed descriptions of the dataset construction process. For the data distribution, we included valid justifications for the distribution.

    (ii) **Evaluation metrics:** We conducted comparative experiments with the reviewer's suggested metrics to demonstrate the validity of our evaluation metrics.

- **To Reviewer A2BT:**

    (i) **Validity of multiple-choice UPD:** We explained that UPD’s multiple-choice format enables more precise evaluation and measures robust understanding capabilities, making it an important step toward assessing LMM reliability.

    (ii) **Benchmark Diversity:** We demonstrated that the MM-UPD benchmark not only matches existing benchmarks in scale but also demonstrates superior functionality through its ability to generate clear and meaningful findings, validating its effectiveness as a UPD evaluation tool. Furthermore, additional experiments with complex datasets like MMMU showed that our MM-UPD Bench enables a more accurate measurement of refusal performance.

- **To Reviewer 48si:** We clearly explained the differences in definitions by referencing the main text. We also provided justification for the table composition.

Finally:

- **Reviewer gvew:** The reviewer’s concerns were addressed, so increased their rating. The reviewer also agreed to Reviewer A2BT that including open-ended questions would enhance the paper's value, but acknowledged the value of this paper.
- **Reviewer A2BT:** The reviewer maintained the score due to the lack of general tasks (e.g., open-ended questions) and their analysis.
- **Reviewer 48si:** The reviewer might have been occupied during the discussion phase and was unable to engage in discussions.

**Important remaining discussion point:**

The key remaining question is whether our work provides sufficient value even if we do not address other problem types, such as open-ended questions (**reviewer A2BT**). While we acknowledge the importance of covering other problem types, we argue that focusing on multiple-choice questions offers distinct advantages: **Our multiple-choice UPD enables more precise evaluation and assesses the robust understanding capability.**  Through our experiments, we have demonstrated important findings to bridge the gap between closed-source LMMs and open-source LMMs. We believe our findings will advance future efforts for reliable LMMs.



We sincerely thank the reviewers for their thorough reviews and the Area Chair for facilitating productive discussions. We hope this summary will assist all reviewers and Area Chairs in reaching their final decision.

---

### Meta-Review · Area_Chair_GxQz · 2024-12-19

**Metareview:**

The paper formulates the problem of unsolvable problem detection (UPD) to evaluate large multimodal models' (LMMs) ability to withhold answers when faced with unsolvable problems. It identifies three distinct settings: Absent Answer Detection (AAD), Incompatible Answer Set Detection (IASD), and Incompatible Visual Question Detection (IVQD). Additionally, the paper introduces the MM-UPD Bench, a benchmark designed to assess UPD performance across various ability dimensions. Through extensive experiments, the paper provides useful insights into developing more reliable LMMs.

**Strengths**

- Expands the scope of existing definitions of unsolvable problems, offering a broader perspective.
- The MM-UPD Bench is a tailored benchmark for evaluating UPD capabilities across multiple ability dimensions.
- Intensive experiments on the MM-UPD Bench provide valuable insights.

**Weaknesses**

- The MM-UPD Bench and experiments are limited to multiple-choice problems, restricting the scope and impact of the work.
- Reviewers expressed concerns about the diversity of the proposed benchmark and lack of comparative analysis regarding data size, distribution, and comprehensiveness.

**Overall Assessment**

The focus on multiple-choice problems limits the generalizability and scope of the proposed work. The exclusion of open-ended questions further narrows the contribution, making it fall below the bar for acceptance. Addressing these limitations in future iterations could enhance the overall impact and utility of the work.

**Additional Comments On Reviewer Discussion:**

While the rebuttal has addressed some of the concerns, major efforts are needed to enhance the scope of the UPD questions and the overall contribution of the work. Expanding the UPD definition and analysis to include open-ended multimodal question-answering scenarios would broaden the impact and applicability of the research.

---

### Decision · Program_Chairs · 2025-01-22

Reject